# Serum and Whole Blood Cu and Zn Status in Predicting Mortality in Lung Cancer Patients

**DOI:** 10.3390/nu13010060

**Published:** 2020-12-27

**Authors:** Katarzyna Zabłocka-Słowińska, Anna Prescha, Sylwia Płaczkowska, Irena Porębska, Monika Kosacka, Konrad Pawełczyk

**Affiliations:** 1Department of Food Science and Dietetics, Wroclaw Medical University, ul. Borowska 211, 50-556 Wroclaw, Poland; anna.prescha@umed.wroc.pl; 2Diagnostics Laboratory for Teaching and Research, Department of Laboratory Diagnostics Wroclaw Medical University, ul. Borowska 211a, 50-556 Wroclaw, Poland; sylwia.placzkowska@umed.wroc.pl; 3Department and Clinic of Pulmonology and Lung Cancers, Wroclaw Medical University, ul. Grabiszynska 105, 53-439 Wroclaw, Poland; irena.porebska@umed.wroc.pl (I.P.); monika.kosacka@umed.wroc.pl (M.K.); 4Department and Clinic of Thoracic Surgery, Wroclaw Medical University, ul. Grabiszynska 105, 53-439 Wroclaw, Poland; konrad.pawelczyk@umed.wroc.pl

**Keywords:** lung cancer, zinc, copper, overall survival

## Abstract

Alterations in circulating Cu and Zn are negative predictors of survival in neoplastic patients and are known during lung cancer. However, no data on predicting mortality of lung cancer patients based on the level of these elements in the blood have been presented to date. The aims of this prospective cohort study were as follows: (i) To evaluate the disturbances in serum and whole blood Cu and Zn, (ii) to assess the relationships between serum and whole blood Cu and Zn status and clinical, sociodemographic, and nutritional data, and (iii) to investigate the association of Cu and Zn status with all-cause mortality in lung cancer. Naïve-treatment lung cancer patients (*n* = 167) were characterized in terms of sociodemographic, clinical, and anthropometric data and dietary intake and compared with sex-matched control subjects (*n* = 48). Whole blood and serum Cu and Zn status was determined by atomic absorption spectrometry. Cox proportional hazards models adjusted for multiple confounders/mediators were used to estimate the association between all-cause death and Cu and Zn status. Sex, cardiovascular disease, chronic obstructive pulmonary disease, clinical stage, and hemoglobin, platelet, and glucose concentrations significantly differentiated Cu and Zn status. All-cause mortality in lung cancer patients was positively associated with serum Cu levels, Cu:Zn ratio, and whole blood Zn levels. However, an advanced clinical stage of disease was the strongest predictor of all-cause mortality. Circulatory status of Cu and Zn might be included in routine clinical characteristics of patients with lung cancer patients as additional prognostic variables, but only after further more detail studies.

## 1. Introduction

Despite many efforts to improve lung cancer outcomes, its prognosis remains poor. Lung cancer is still a leading cause of cancer-related death, with the highest mortality rate in the world. Moreover, the morbidity of lung cancer classifies this disease as having the highest prevalence among cancers, especially in developed countries [1]. Although screening programs enable diagnosis at early stages, thus reducing the mortality rate [2], they are still unavailable routinely [3]. This leads to lung cancer diagnosis at advanced clinical stages, with severely limited treatment options and poor five-year survival. The clinical stage of disease is one of the most important but unmodifiable factors related to prognosis; therefore, the identification and availability of prognostic markers other than the stage are important for more precise detection of cancer progression, and the initiation of personalized treatment regimens [4].

Altered trace element status during cancerogenesis and disease progression is widely known and considered as a cause or effect of cancerogenesis, depending on the type of trace element, the cancer, and many other factors [5,6,7]. Such alteration has been observed in lung cancer related to abnormal concentrations of serum Zn and Cu [8,9], indicating that disturbances in serum Cu–Zn homeostasis influence lung cancerogenesis [10,11]. However, no studies on whole blood Cu–Zn homeostasis in lung cancer have been performed before, except ours [12]. This indicated that disturbed whole blood Cu, Zn, and the Cu:Zn ratio were not reflected in alterations presented in serum, suggesting an additional between-compartment disorder. Cu and Zn are interdependent trace elements involved in numerous aspects of cellular metabolism and immune function, and are co-factors of numerous enzymes that have redox and other activity [13]. The Cu–Zn balance is often disturbed during systemic breakdown of whole body homeostasis due to, e.g., malabsorption, excessive excretion, abnormal requirements, inflammatory processes, and redox imbalance related to chronic illness, e.g., cancer [13,14]. Additionally, demand for Cu is elevated in proliferating cancer cells relative to other tissues; thus, metabolic vulnerability can be exploited by controlling Cu availability during cancer progression [15]. Although mortality prediction based on serum Cu and Zn status has been studied in cancers [16,17], to date no data on the role of Cu and Zn disturbances in predicting the mortality of lung cancer patients have been presented. Moreover, no studies on the prediction of mortality based on whole blood Cu and Zn levels and the Cu:Zn ratio have been performed in lung and other cancers.

The aims of this prospective cohort study were as follows: (i) To evaluate the disturbances in serum and whole Cu and Zn status, (ii) to assess the relationships between serum and whole blood Cu and Zn status and clinical, sociodemographic, and nutritional data, and (iii) to investigate the association between serum and whole blood Cu and Zn status and all-cause mortality among patients with lung cancer.

## 2. Materials and Methods

### 2.1. Study Design and Endpoints

This study was designed to evaluate the following hypotheses: (i) Circulating (serum and whole blood) Cu and Zn status is disturbed in lung cancer patients and (ii) this alteration is related to the following parameters; clinical: Stage of disease, prevalence of co-morbidities like cardiovascular disease (CVD), chronic obstructive pulmonary disease (COPD), diabetes mellitus (DM) type 1 or 2, anemia prevalence, alterations in hemoglobin (Hb), platelets, serum creatinine, glucose concentration, estimated globular filtration (eGFR), and neutrophil:lymphocyte ratio (NLR); nutritional: Nutritional status (with emphasis on overweight and obesity prevalence), dietary intakes of total energy, macronutrients, and microelements; sociodemographic: Sex, age, level of education, active and passive smoking status as well as alcohol consumption, and (iii) serum and whole blood Cu and Zn can be used to predict all-cause lung cancer death (endpoint), since alterations in circulatory Cu and Zn status as well as the above-mentioned factors related to clinical, nutritional, and sociodemographic data influence all-cause mortality among lung cancer patients.

### 2.2. Study Subjects

Newly diagnosed, naïve-treatment lung cancer patients (*n* = 167) were enrolled in this study. All patients were recruited during the first day hospital admission, prior to any oncological treatment regimens. Patients were recruited from the Lower Silesian Centre of Lung Diseases between June 2013 and December 2016. Inclusion criteria for lung cancer patients were as follows: Confirmed diagnosis of primary lung cancer, lack of mental disease which could make it difficult to obtain reliable answers, no other primary malignant diseases. All lung cancer diagnoses were confirmed by histopathological examination after surgery or bronchofiberoscopy. The clinical stage of disease and metastases were evaluated based on chest computed tomography (CT), positron emission tomography (PET)-CT, and ultrasonography of the abdominal cavity. CT/MRI of the central nervous system and scintigraphy of bone were performed if necessary. In cases of enlarged lymph nodes of the mediastinum, endobronchial ultrasound transbronchial needle aspiration (EBUS-TBNA) was performed. Negative EBUS results were verified by mediastinoscopy. Exclusion criteria for lung cancer patients were as follows: Unconfirmed diagnosis of lung cancer, lung cancer as metastasis of other primary cancer, prior chemotherapy, radiotherapy or other oncological therapy for any malignant disease.

The control group (*n* = 48) consisted of healthy people recruited from the Universities of the Third Age in Wroclaw and public offices. Exclusion criteria for the control group were as follows: Cancers, metabolic disturbances, other pro-inflammatory diseases, and mental diseases. The patient and control groups were sex matched. The control group was characterized analogously as lung cancer patients except for their clinical characteristics.

The study protocol was approved by the Ethics Commission of Wroclaw Medical University (approval No. 540/2013), and the study was conducted in accordance with the principles expressed in the Declaration of Helsinki. All participants provided written consent to take part in the research.

### 2.3. Sociodemographic Characteristics

Sociodemographic characteristics of lung cancer patients and control subjects were determined based on the following criteria: Sex, age, level of education, smoking and alcohol drinking status. In terms of smoking, all participants were classified as never (never or >1 year smoking cessation), previous (≤1 year smoking cessation), and current (active smoking or ≤2 weeks smoking cessation). The number of cigarettes was assessed only for current smokers. Participants were classified as alcohol consumers (at least 10 g of ethanol/month) or non-consumers.

### 2.4. Clinical Characteristics

Patients were characterized in terms of clinical stage of disease, type of lung cancer, type of oncological treatment, and the presence of the following concomitant diseases: Diabetes mellitus (DM) type 1 or 2, cardiovascular diseases (CVD), or chronic obstructive pulmonary disease (COPD). Laboratory analysis was conducted to assess the clinical condition of lung cancer patients: Whole blood concentration of hemoglobin, platelets count, and neutrophil:lymphocyte ratio (NLR) were determined by a Sysmex IXP 1080i automatic hematology analyzer; serum concentration of creatinine and glucose and estimated glomerular filtration rate (eGFR) calculation were performed with a Cobas Roche Integra 400+. All biochemical analyses were performed routinely in the hospital. Anemia was diagnosed as Hb concentration <13.7 g/dL in men under 60 years old, <13.2 g/dL in men over 60 years old, and <12.2 g/dL in all women, regardless of age [18]. The reference range of platelet count was 150–400 × 10^3^ cells/µL [19]. Patients were divided into 3 groups in terms of fasting glucose concentration: Normal (<100 mg/dL), impaired (100–125 mg/dL), and diabetic (≥126 mg/dL) [20]. The normal eGFR rate was defined as ≥90 mL/min/1.73 m^2^, while the normal range of serum creatinine concentration was 0.7–1.2 mg/dL [21].

### 2.5. Anthropometric Measurements and Dietary Intake

Due to the significant impact of nutritional status on circulating Zn and Cu status, we performed baseline anthropometric measurements to evaluate these associations among participants enrolled in this study, with emphasis on overweight and obesity [22,23]. Anthropometric parameters as well as energy and nutrient intake were used to assess the nutritional status of participants. Baseline anthropometric parameters were as follows: Weight (kg), height (cm), body mass index (BMI; kg/m^2^), body fat percentage (BFP; %), and waist–hip ratio (WHR; arbitrary unit). The percentage of body fat was determined using a body fat monitor (Omron BF 306, Osaka, Japan). WHR was calculated as the ratio of waist to hip circumference. Waist circumference was measured at the level of the umbilicus, and hip circumference at the trochanter. Abdominal obesity was determined based on WHR, and following WHO criteria: ≥0.90 for men, ≥0.85 for women [24]. All anthropometric measurements were performed twice and a mean was used for further analysis.

Dietary data on intake of energy, macronutrients (total proteins, fats, and carbohydrates), dietary fiber, and selected minerals based on 3-day dietary recall were collected from subjects by a trained interviewer and calculated with Dieta 6.0 (National Food and Nutrition Institute, Warsaw, Poland) and checked by one independent professional. The program automatically allowed for the reduction of nutrients through technological loss in order to obtain net worth. In the case of lung cancer patients, collection of dietary data took place on the first day of hospital admission and concerned the last three days prior to admission to hospital. All study participants were asked about all food products, dishes, and drinks consumed in the last three days. To assess information about the portion sizes of food products, dishes, and drinks, and to avoid differences in size estimation, the “Album of Photographs of Food Products and Dishes” (National Food and Nutrition Institute, Warsaw, Poland) was used. All participants and subjects had access to “Album of Photographs of Food Products and Dishes” to indicate portion sizes.

### 2.6. Blood Sample Collection

Blood samples from lung cancer patients and control subjects were collected after overnight fasting. In the case of patients, samples were taken the day after hospital admission. Serum was separated by centrifugation after at least 30 min of clotting at room temperature. Aliquots of whole blood and serum were stored at −80 °C until analysis.

### 2.7. Whole Blood and Serum Zn and Cu Determination

Frozen whole blood was thawed at room temperature and then vortexed until completely homogenized. A 0.5-mL sample was mineralized by microwave with a wet mineralization technique in a closed system using an MLS 1200 Mega oven (Milestone, Bergamo, Italy) with 6 mL of a 1:5 mixture of H_2_O_2_, 30% (Sigma Aldrich, St. Louis, Missouri, USA) and HNO_3_, 69–70% (Baker Chemicals, Phillipsburg, NJ, USA). Each sample was mineralized in duplicate and then diluted 20 times with deionized water. Frozen serum was thawed at room temperature and then vortexed until completely homogenized (~5 s). Then the serum sample (0.4 mL) was diluted 1:5 with deionized water. Whole blood Zn, Cu, and serum Zn concentrations were determined by atomic absorption spectrometry with flame atomization (F-AAS) and whole blood Cu concentration with graphite furnace atomization (GF-AAS) using a PinAAcle™ 900T spectrometer (Perkin Elmer, Waltham, MA, USA). For the determination of serum and whole blood Zn concentrations, a Zn hollow cathode lamp at 213.86 nm wavelength was used, and for serum Cu, a Cu hollow cathode lamp at 324.75 nm. For the determination of both trace elements, a slit width of 0.7 nm and an air (10 L/min) acetylene (2.5 L/min) burner were used.

Data on the GF-AAS technique for whole blood Cu were as follows: Wavelength of hollow cathode lamp, 324.75 nm; pyrolysis temperature, 1300 °C; atomization temperature, 2200° C; slit width, 0.7 nm; inert gas, argon; palladium–magnesium nitrate modifier, 5 μL; and sample volume, 20 μL. The accuracy of the method was measured with certified reference materials (Seronorm TM Trace Elements Whole Blood L-3, Sero AS, Billingstad, Norway). Analytical values of serum Zn and Cu in the reference materials were 2520 and 2887 µg/L, respectively, and of whole blood Zn and Cu were 9.11 and 2.47 mg/L, respectively. Mean accuracy (*n* = 6) was as follows: Whole blood Zn, 98.8%; whole blood Cu, 106.6%; serum Zn, 102.2%; and serum Cu, 105.1%. Measurement of serum Zn and Cu concentration was repeated 4 times for each sample, and measurement of whole blood was repeated 3 times for each of 2 sample preparations (6 measurements for each whole blood sample). Repeated analyses were accepted when relative standard deviation (RSD) was less than 5%; however, most analyses gave RSD within 1–2%.

### 2.8. Statistical Analysis

All data were presented as median values (Q1–Q3). The data were analyzed using Statistica 13, PL (Statsoft, Tulsa, Oklahoma, USA). We used the Shapiro–Wilk test to analyze the normality of data distribution within groups and extracted subgroups. To evaluate the differences in the distribution of parameters related to Cu and Zn status, as well as continuous clinical, sociodemographic, and nutritional data between lung cancer patients and the control group, the Student’s *t*-test (for parametric data distribution) or the Mann–Whitney U test (for nonparametric data distribution) were performed. A chi-squared test was used to determine significant differences in the distribution of categorical variables of sociodemographic, nutritional, and clinical characteristics between groups.

We followed the lung cancer study population from the date of diagnosis until death or 1 September 2020, with a median (range) follow-up time for this study of 43.04 (0.23–85.81) months. Death was established based on data from the Registry Office. Lung cancer patients were linked to the data statistics registry records using a unique 11 digit identification number (PESEL), first name, and last name. Since the cause of death was not available, only all-cause mortality was analyzed.

We evaluated the relationship between serum and whole blood Cu and Zn concentration, Cu:Zn ratio, and all-cause death using Cox proportional hazard analysis. First, we built univariate Cox regression models to evaluate the significance in predicting mortality for each parameter related to clinical (categorical variables), sociodemographic (categorical variables), and anthropometric (categorical variables) data and for trace element status (continuous variables): Serum Cu per 0.1 mg/L, serum Zn per 0.1 mg/L, serum Cu:Zn ratio per 0.1 arbitrary unit, whole blood Cu per 0.1 mg/L, whole blood Zn per 1 mg/L, and whole blood Cu:Zn ratio per 0.01 arbitrary unit. Then, for each parameter related to the trace elements (serum and whole blood Cu, Zn, Cu:Zn ratio), we built a model of multivariate Cox regression analysis. Models I, II, and IV–VI were adjusted by age (continuous), sex, passive smoking (yes/no), smoking status (yes/no/previous), alcohol consumption (yes/no), BMI (<20/20–24.9/>24.9 kg/m^2^), abdominal obesity (yes/no), DM (yes/no), CVD (yes/no), COPD (yes/no), clinical stage (I, II, III, IV), anemia (yes/no), platelet count (<150/150–400/>400 10^3^ cells/µL), and NLR (<2.36 vs. >2.36 arbitrary unit; median NLR ratio was 2.36 for lung cancer patient group). Model III was adjusted by age (continuous), sex, passive smoking (yes/no), smoking status (yes/no/previous), alcohol consumption (yes/no), BMI (<20/20–24.9/>24.9 kg/m^2^), DM (yes/no), CVD (yes/no), COPD (yes/no), anemia (yes/no), and platelet count (<150/150–400/>400 10^3^ cells/µL). The stability of the model was certified by using a likelihood ratio step-forward fitting procedure. Additionally, overall 86-month survival according to serum and whole blood Cu and Zn levels and Cu:Zn ratio was demonstrated using Kaplan–Meier curves. The relationship between each of these variables (presented as categorical data: < and ≥ median of concentration in whole lung cancer patient group) and overall survival were assessed by log rank test. For all statistical procedures, the significance level was considered to be <0.05.

Receiver operating characteristic (ROC) curves, area under the ROC curves (AUCs), and 95% confidence intervals for AUCs as well as Youden’s index were calculated to determine cut-off values of trace elements, which best differentiated survival vs. non-survival lung cancer patients, as well as clinical stage IV vs. clinical stages I–III disease.

## 3. Results

### 3.1. Baseline Characteristic

Baseline characteristics of lung cancer patients and the control group are presented in Table 1. Lung cancer patients were significantly older than the control group and a significantly lower percentage never smoked (26.5 vs. 66.7%), consumed alcohol regularly (49.0 vs. 84.4%), or consumed less alcohol. The two groups did not differ in terms of sex, education level, passive smoking, and number of cigarettes smoked. Anthropometric parameters revealed that lung cancer patients weighed significantly less (71.0 vs. 78.0 kg) and had a significantly lower percentage of adipose tissue (27.8 vs. 33.1%), but more often presented abdominal obesity (77.1% vs. 50.0%). Nutritional data indicated that patients and control subjects did not differ in terms of energy, macronutrient, and microelement intake, except for significantly higher calcium intake (743.1 vs. 571.1 mg/d). Lung cancer patients were diagnosed and enrolled in this study at different clinical stages of disease. A majority of them had clinical stage I and II disease and 21.3% had metastasis (clinical stage IV disease). Post-study oncological treatment was as follows: Two-thirds of patients underwent lung cancer resection, 40.7% of subjects received chemotherapy, and 10.8% received radiotherapy. The majority of lung cancer participants suffered from non-small cell lung cancer. Among all patients, about 53% had been diagnosed with at least one of the following diseases: CVD, COPD, or DM type 1 or 2. In detail, about half suffered from CVD, about 14% from COPD, and 15% had DM type 1 or 2. Additionally, about 40% of lung cancer subjects had iron-deficiency anemia. Reactive thrombocytosis, diagnosed as elevated platelet count (>400 × 10^3^ cells/µL), was observed in every thirteenth lung cancer patient; on the other hand, about 8% of lung cancer patients suffered from thrombocytopenia (<150 × 10^3^ cells/µL). One-third of patients presented decreased eGFR (<90 mL/min/1.73 m^2^) and a similar proportion had serum creatinine levels <0.7 mg/dL, which may indicate abnormal renal function. More than half of the patients had elevated serum glucose concentration (≥100 mg/dL), while every seventh lung cancer subject had serum glucose concentrations indicating diabetes mellitus.

### 3.2. Serum and Whole blood Cu and Zn Status in Lung Cancer and Its Relationships with Clinical, Sociodemographic, and Nutritional Data

Serum and whole blood Cu and Zn concentrations and Cu:Zn ratio in lung cancer and control subjects and subgroups of lung cancer patients extracted according to clinical, anthropometric, and sociodemographic factors are presented in Table 2. In lung cancer patients compared to control subjects, the median serum Zn concentration was significantly lower: 0.88 vs. 1.23 mg/L, while the serum Cu:Zn ratio was significantly higher: 1.18 vs. 0.83, respectively. Similar trends were observed in whole blood. Importantly, the median whole blood Cu:Zn ratio in the lung cancer group was twice the value of the control group. Moreover, whole blood Cu levels were significantly increased in lung cancer patients compared to control subjects. Sex significantly differentiated the serum Cu concentration, Cu:Zn ratio, and whole blood Cu:Zn ratio. Women with lung cancer had significantly higher serum Cu concentration, serum Cu:Zn ratio, and whole blood Cu:Zn concentration than men. Interestingly, the presence of CVD or COPD was associated with higher whole blood Zn concentration and lower whole blood Cu:Zn ratio in lung cancer patients. Metastasis (clinical stage IV) significantly altered serum and whole blood concentrations of analyzed trace elements: Serum Cu and whole blood Zn levels were significantly higher compared to patients with clinical stage I disease. Cut-off values of serum Cu and whole blood Zn concentrations that best differentiated patients with clinical stage IV vs. clinical stage I–III were as follows: 1.42 and 7.12 mg/L, respectively, with following AUCs (95% CI): 0.658 (0.537–0.779) and 0.705 (0.599–0.811), respectively (both *p* < 0.05) (Appendix A).

Patients with decreased Hb concentration had significantly higher serum Cu levels and a Cu:Zn ratio, while serum Zn levels were significantly lower. Moreover, increased platelet count was associated with significantly higher serum Cu concentration compared to patients with normal or decreased platelet counts. Additionally, an abnormal glucose concentration >125 mg/dL was associated with significantly higher whole blood Zn level. Ca intake below estimated average requirements was associated with significantly higher serum Cu concentration in lung cancer patients. The presence of DM, NLR concentrations (below vs. above median concentration for lung cancer group), eGFR (below vs. above reference range), creatinine (below vs. above reference range), age (below vs. above 60 years old), smoking status (yes vs. no vs. previous), alcohol consumption (yes vs. no), and anthropometric parameters: BMI (<20 vs. 20–24.9 vs. >25 kg/m^2^) and WHR (abdominal obesity: Yes vs. no) did not influence serum and whole blood concentrations of Cu and Zn or the Cu:Zn ratio (Appendix A).

### 3.3. Associations between Serum and Whole Blood Cu and Zn Status and All-Cause Mortality Among Patients with Lung Cancer

Table 3 presents significant factors predicting all-cause mortality in lung cancer patients. In the univariate Cox analysis, smoking status, clinical stage III and IV disease, NLR above the median of the studied group, platelets >400 × 10^3^ cells/µL, glucose >125 mg/dL, increased serum Cu concentration per 0.1 mg/L, Cu:Zn ratio per 0.1 arbitrary unit, and, surprisingly, increased whole blood Zn concentration per 1 mg/L were all significant predictors of overall mortality among lung cancer patients. The strongest predictive factor was an advanced clinical stage of disease; patients with stage IV had a 4.5 times higher hazard ratio (HR) than patients with stage I disease. Among parameters related to Cu and Zn status, whole blood Zn concentration was the strongest factor predicting mortality, followed by serum Cu concentration and serum Cu:Zn ratio. Increased whole blood Zn per 1 mg/L was significantly associated with a 32% higher HR for all-cause mortality, serum Cu per 0.1 mg/L with 18%, and Cu:Zn ratio per 0.1 arbitrary unit with 6%.

Table 4 presents six models of multiple Cox regression analysis concerning mortality prediction in the lung cancer group. The models revealed that elevated serum Cu concentration, Cu:Zn ratio, and whole blood Cu and Zn concentrations significantly increased the HR of all-cause mortality. Serum Zn concentration and whole blood Cu:Zn ratio did not predict mortality in multiple models. Moreover, clinical stages III and IV were more strongly associated with mortality (HR = 2.59 for stage IV in model V to HR = 3.81 for stage III in model IV) than levels of particular factors related to Cu and Zn (HR = 1.06 for increased serum Cu:Zn ratio per 0.1 arbitrary unit to HR = 1.25 for increased whole blood Zn concentration per 1 mg/L). Additionally, in model III concerning the prediction of all-cause death using whole blood Cu concentration, DM prevalence also strongly predicted the risk of death (HR = 1.83).

Kaplan–Meyer overall survival (OS) estimates according to median serum Cu concentration, Cu:Zn ratio, and whole blood Zn concentration are presented in Figure 1, Figure 2 and Figure 3. Serum Cu concentrations below the median had a significant beneficial effect, increasing the probability of 5-year OS from ~40% to 59% (Figure 1). Similarly, a low serum Cu:Zn ratio significantly increased the probability of 5-year OS from 41% to 59% when compared with serum Cu:Zn ratio above median (Figure 2). Interestingly, whole blood Zn below the median was significantly associated with higher 5-year OS: 56% probability vs. 43% for whole blood Zn above the median (Figure 3).

We estimated the cut-off values—1.19 mg/L for serum Cu concentration, 1.34 for serum Cu:Zn ratio, and 7.54 mg/L for whole blood Zn concentration‚which best differentiated survival from non-survival lung cancer patients. Areas under the ROC curve (AUC, 95% CI) for serum Cu concentration, serum Cu:Zn ratio, and whole blood Zn concentration were as follows: 0.668 (0.586–0.750), 0.604 (0.518–0.690), and 0.637 (0.551–0.723) (Appendix A). Follow-up times and events of all-cause cancer death in lung cancer patients by serum Cu, whole blood Zn, and serum Cu:Zn ratio are presented in Table 5. Serum Cu concentration above the median displayed the highest incidence rate: 16.53 per 1000 person months, while serum Cu below the median had the lowest incidence rate with 8.45 per 1000 person months. The medians of serum Zn concentration, whole blood Cu concentration, and whole blood Cu:Zn ratio did not differentiate OS in the lung cancer group (Appendix A).

## 4. Discussion

This study, to our knowledge, is the first to examine the association of circulating (serum and whole blood) Cu and Zn status with overall lung cancer survival. Data on associations between mortality of cancer patients and these trace elements refer to dietary intake rather than body status. Lower dietary Zn intake was significantly associated with higher all-cause mortality including lung, prostatic, and esophageal cancer; additionally, Cu intake was also inversely related to mortality of esophageal cancer patients [25,26,27]. In a study by Ito et al. [16] on all-cause cancers, whole blood Zn and Cu status were both associated with mortality. The risk of mortality increased by almost double in patients with the lowest values of blood Zn:Cu compared with those with the highest values. Other investigations concerning Cu and Zn homeostasis and survival prediction were conducted mainly among elderly people, where the plasma Cu:Zn ratio was considered as an important clinical inflammatory-nutritional biomarker and significant predictor of all-cause mortality in those over 70 years old [28].

In our prospective cohort study, we found that Cu and Zn status in lung cancer was disrupted mainly via increased circulatory Cu with concomitant decreased Zn concentration. This was reflected in a significantly higher serum and whole blood Cu:Zn ratio in lung cancer patients compared to control subjects. However, the demonstrated alterations were not associated with the patient’s age, as was presented by Malavolta et al. [28]. On the other hand, we found, in line with other authors [28], that women had higher circulating Cu concentration and Cu:Zn ratio than men. Results observed in this study concerning abnormal Cu and Zn status are in agreement with previous findings, suggesting a markedly elevated circulating Cu:Zn ratio in breast, hepatobiliary, colorectal, bladder, and other cancers [5,29,30,31]. Further studies are needed to more precisely evaluate disturbances in circulating Cu in lung cancer, mainly via assessment of the relative levels of free- and ceruloplasmin-bound Cu and their influence on cancerogenesis and mortality. Unbound-Cu is responsible for free radical reactions, e.g., the Haber–Weiss reaction, and thus lead to alterations in functioning of key biomolecules as protein, lipids, and nucleic acids [32].

As co-factors of many enzymes, including the key Cu/Zn enzyme superoxide dismutase, these trace elements play critical roles in maintaining DNA stability and integrity by eliminating superoxide anions to prevent cancerogenesis [33]. Therefore, an abnormal circulating Cu and Zn concentration may contribute to cancer development by dysregulation of redox enzyme activity, and therefore DNA damage and protein modification [34]. On the other hand, higher Cu concentration is needed to meet the metabolic demands of the growth and development of cancer cells. Cancer cells require higher Cu resources for e.g., the Warburg effect—an early metabolic switch by which cancer cells produce energy via glycolysis instead of mitochondrial oxidative phosphorylation. Moreover, increased Cu concentration is needed to potentiate the activity of cupro-enzymes like lysyl oxidase (LOX), superoxide dismutase (SOD), and cytochrome-c-oxidase (COX) as well as some molecular pathways which support cancer cell proliferation and survival, like mitogen-activated protein (MAP) kinases and autophagic ULK1/2 kinases [15]. Additionally, Cu and Zn have specific and different activities. Cu is mainly involved in redox homeostasis and inflammatory processes, while Zn maintains cell replication, growth, and repair as well as nucleic acid metabolism and proper immune function [35]. Therefore, an increased circulating Cu:Zn ratio has been linked with homeostasis breakdown toward systemic inflammation, and was even proposed as an independent biomarker of this phenomenon [36]. The host inflammatory response during cancerogenesis has been proven to be significantly associated with the risk and extent of metastatic involvement in many cancer types, including lung cancer [37,38]. Inflammation might play a key role in promoting cancer progression and metastasis, because its mediators increase vascular permeability, favor cancer cell infiltration, and contribute to cancer cell adhesion to endothelium and stromal invasion at metastatic sites [39].

In this study, we observed that patients with metastasis showed higher circulating Cu concentrations and a tendency for a higher serum Cu:Zn ratio. Altered compartmentalization of these two trace elements during cancerogenesis contributes to increased circulatory Cu levels, and a declining Zn concentration is related to cancer inflammatory processes and progression, irrespective of diet. This is in agreement with results presented in this study, where lung cancer patients did not differ from control subjects in terms of Cu and Zn intake. As circulating Cu:Zn ratio increased significantly in advanced stages [34,35,40], this biomarker may be a reliable prognostic factor for mortality in neoplasms, including lung cancer. Indeed, our further analysis indicated that the serum Cu:Zn ratio is a good prognostic marker of all-cause mortality in lung cancer, indicating that elevated serum Cu:Zn per 0.1 mg/L increases the risk of all-cause death by 6%, and increased serum Cu concentration per 0.1 mg/L by 13%. Cu is known to play an integral role in metastasis and is involved in energy homeostasis regulation, the epithelial–mesenchymal transition (EMT) in cancer, and cell migration and invasion [41,42]. Additionally, Cu is considered as a “switch-on” proangiogenic messenger due to its critical role as a co-factor of proangiogenic compounds [43]. It has been shown that expression of VEGF, one of the crucial components of tumor angiogenesis, is Cu-mediated, partly by activation of hypoxia-inducible factor 1-α [44,45]. Moreover, FGF1, angiogenic growth factor, is a Cu-binding protein and its secretion requires the Cu-dependent formation of a complex. Cu also potentiates the affinity with which angiogenin, and binds to endothelial receptors. Additionally, antioxidant-1 (Atox-1), cytosolic protein, is responsible for transferring Cu(I) to the Cu-ATPases in inflammatory neovascularization via regulating the ROS/NF-κB pathway [44]. Therefore, an increase of Cu might negatively affect survival prognosis, indicated in this study, also via potentiated angiogenesis.

In our study, we also found interesting and surprising results concerning the association between whole blood Zn status and all-cause mortality among lung cancer patients. Increased whole blood Zn per 1 mg/L was associated with a 25% increase of all-cause mortality among lung cancer patients. This is the first study concerning whole blood Zn status in patients with cancer as a predicting factor of all-cause mortality; therefore, no reference can be made to other studies. This result, however, needs further research in a larger, more homogeneous cohort (probably with advanced clinical stage of disease), due to inconsistency with data, indicating that Zn has rather anti-cancerogenic, anti-inflammatory, and anti-metastatic activity [46]. The association observed in this study could have resulted from altered compartmentalization of Zn, especially in patients with metastasis. As mentioned above, until now, there have been no studies concerning the changes in between-compartment homeostasis of these trace elements in lung cancer except ours [12]; therefore, studies exploring the mechanism of this perturbation in more detail should be performed.

An appropriate Zn level is needed for homeostasis in cellular function, such as preventing apoptosis in normal cells and promoting apoptosis when it is advantageous to the organism [47]. Therefore, inhibition of apoptosis during the cancerogenic process could result from Zn insufficiency rather than elevation of circulating Zn. Indeed, based on several other studies [29,48,49], circulating Zn concentration decreases during cancerogenesis. Moreover, Zn deficiency is involved in several stages of malignant transformation [50]. In this study, we found that serum and whole blood Zn in the lung cancer group decreased compared to control subjects; however, surprisingly, it increased in whole blood during the progression of disease. Data on the results concerning abnormal whole blood Zn levels in cancer patients are not consistent. In a study by Aldor et al. [48], the concentration of whole blood Zn in patients with cancer was significantly lower compared to control subjects. A similar trend of whole blood Zn status in women with gynecological and breast cancer was shown by Memon et al. [49]. However, Kucharzewski et al. [51] found that whole blood Zn levels in patients with thyroid cancer were significantly higher than in control subjects. It should be noted that in presented above studies, associations between stage of disease and whole blood Zn status were not evaluated. Data on serum Zn status were more consistent and indicated that serum Zn concentration was lower in oncological patients than in healthy subjects [9,31].

Apart from the clinical stage of disease, we also evaluated other clinical data to find the impact of disruption to the homeostasis of the analyzed trace elements. A higher Cu concentration and Cu:Zn ratio were associated with anemia prevalence and elevated platelet count, and these were mainly observed in serum. Cancer-related anemia was characterized by biological and hematologic features that resemble those described in anemia associated with chronic inflammatory disease [52]. Therefore, the positive relationship between circulating Cu concentration and anemia in this study probably mainly resulted from a common link, the inflammatory process, and were often observed in lung cancer patients ceruloplasmin alterations. Ceruloplasmin plays a crucial role in iron release from cells and tissue into circulatory system. Patients with ceruloplasmin deficiency may accumulate iron in tissue and therefore their serum iron concentration is decreased [53]. However, we did not evaluate serum ceruloplasmin concentration in this study; therefore, alterations in ceruloplasmin that are mentioned above remain only speculative, and require further research. Among lung cancer patients enrolled in this study, anemia was not associated with increased mortality. Similar to anemia, a mechanism concerning the relationship between Cu status and platelet count might be proposed. Platelets serve various roles in pathological pathways, including inflammation, by secreting proangiogenic cytokines [54]. However, platelets are also directly linked to oncological processes [55]. Thrombocytes may augment tumor growth and enhance metastasis by supporting angiogenesis, protecting tumor cells from clearance by the immune system by accumulating on circulating embolic tumor cells. On the other hand, increased platelet counts may be an epiphenomenon of tumor growth, as tumor-secreted cytokines can induce thrombopoiesis [55,56]. Therefore, increased serum Cu along with increased platelet count might result from inflammatory processes and/or disease progression. Although elevated platelet count is related to cancer progression and therefore overall survival, as demonstrated in several studies, including in lung cancer [57,58,59], we did not indicate that alterations in platelet count contributed to increased mortality in the studied group.

We have shown that CVD and COPD prevalence substantially increased whole blood Zn concentration. Although results presented elsewhere [60,61] indicated that circulating Zn concentration instead decreased in the presence of CVD or COPD, there are no data on the relationship of whole blood Zn concentration with these morbidities. In addition, there are no data on circulatory Zn concentration in lung cancer patients with COPD and/or CVD as comorbidities. Therefore, further in-depth research should be conducted to evaluate body Zn status among patients with several serious morbidities. Additionally, serum glucose concentration >125 mg/dL was associated with higher whole blood Zn level and was an independent risk factor of all-cause mortality among our patients. In Luo et al.’s study [62], elevated fasting blood glucose was also independently associated with a substantially higher risk of all-cause mortality in lung cancer patients. This indicates that effectively controlled diabetes or hyperglycemia may improve survival among lung cancer patients with abnormal glucose levels [62].

Our study showed for the first time the association of serum, whole blood Zn and Cu levels, and Cu:Zn ratio with lung cancer prognosis in naïve-treatment patients, taking into consideration a multifaceted background including clinical stage, comorbidities, clinical parameters, smoking, alcohol, and nutritional status. Moreover, this study was performed with a long-term follow-up of 86 months. An additional advantage of this study is that it estimated the cut-off values of trace element concentrations significantly affecting mortality in the studied group of lung cancer patients. The present study also had several limitations. First, it was a prospective cohort study performed in a single center with a relatively small number of patients; therefore, the present results may not be representative of the general population of lung cancer patients, and further studies are warranted to confirm these results. Moreover, we did not perform measurements of ceruloplasmin concentration; therefore, we were not able to assess ceruloplasmin-bound and ceruloplasmin-free Cu. This information is warranted for further evaluation of mechanisms related to associations between Cu status and risk of mortality in lung cancer patients as well as lung cancer-related anemia. In addition, there is a risk of residual confounding due to the use of self-reported information (e.g., smoking status, alcohol consumption). Moreover, although we performed our analysis using multivariable Cox proportional hazard analysis, we cannot rule out other residual confounding variables and that the obtained effects were not completely adjusted by other factors. Actually, circulating Cu and Zn status has been shown to be associated with dietary intakes, disturbances in redox status, immune system, and metabolic derangements [12,15,46,63]. In this study, it was however not possible to control all factors influencing Cu and Zn status such as diet, lifestyle, and clinical parameters; therefore, further studies should be performed in a more homogenous group of lung cancer patients, or analyses should also be adjusted for the above-mentioned data to better associate the impact of circulating Cu and Zn status on predicting mortality in lung cancer patients.

## 5. Conclusions

In conclusion, a higher serum Cu level, Cu:Zn ratio, and whole blood Zn level were shown to negatively affect all-cause mortality risk in lung cancer patients. However, further, more detailed, and with a larger cohort studies on the role of Cu and Zn status in predicting all-cause and cancer-cause mortality among lung cancer patients as well as studies on mechanisms underpinning surprising negative prediction of lung cancer survival in patients with elevated whole blood Zn status are warranted. This could be helpful in developing strategies to improve the outcome of patients with regard to the body status of these elements. Findings of the study indicate that circulatory status of Cu and Zn might be included in routine clinical characteristics of patients with lung cancer patients as additional prognostic variables, but only after the further above-mentioned studies.

## Figures and Tables

**Figure 1 nutrients-13-00060-f001:**
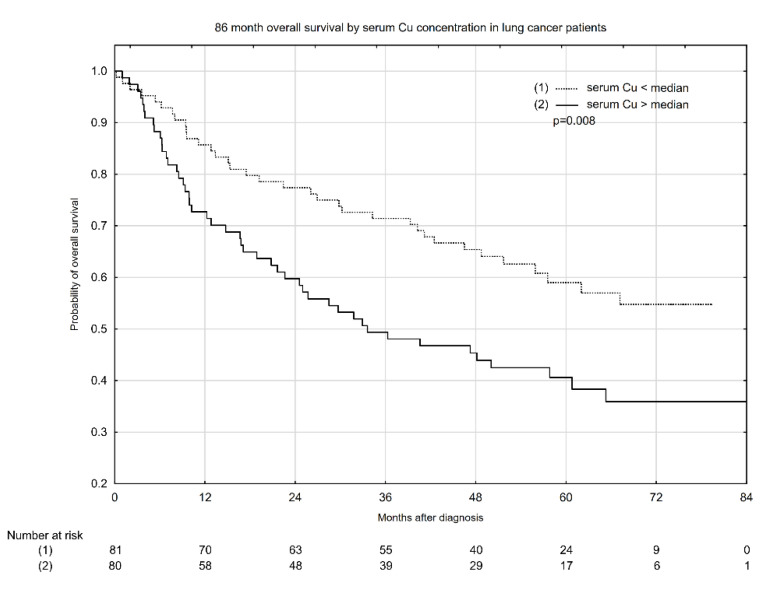
Eighty-six-month overall survival by serum Cu concentration in lung cancer patients.

**Figure 2 nutrients-13-00060-f002:**
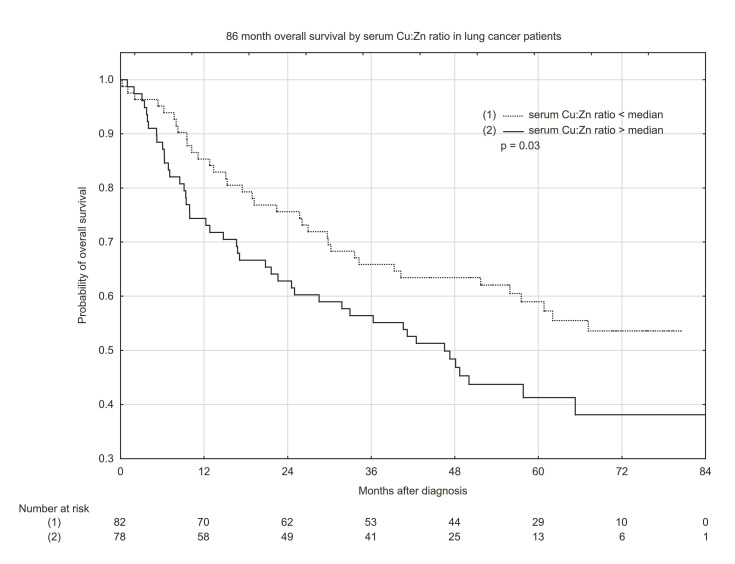
Eighty-six-month overall survival by serum Cu:Zn ratio in lung cancer patients.

**Figure 3 nutrients-13-00060-f003:**
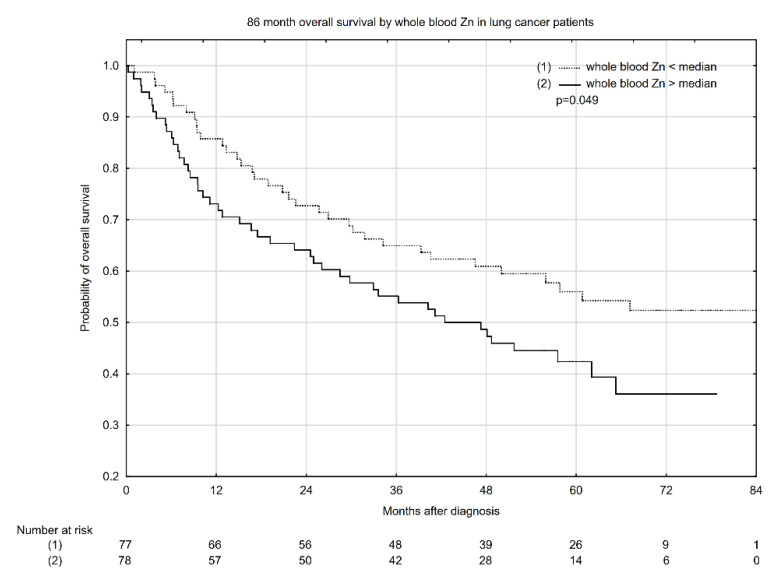
Eighty-six-month overall survival by whole blood Zn concentration in lung cancer patients.

**Table 1 nutrients-13-00060-t001:** Baseline characteristic of lung cancer patients and control group.

Parameter	Lung Cancer Patients	Control Group	*p*
Sociodemographic Factors			
**Sex**, **F/M** (**%, *n***)	45.8/54.2 (76/90)	54.2/45.8 (26/22)	0.306
**Age**, **years** (**median (Q**–**Q3)**)	66.0 (60.0–73.0)	58.0 (45.0–66.0)	<0.001
**Education**: **Primary/vocational/high school/college** (**%, *n***)	16.3/32.7/34.6/16.3 (25/50/53/25)	6.7/22.2/42.2/28.9 (3/10/19/13)	0.118
**Smoking status: Never/previous/current** (**%, *n***)	26.5/45.2/28.4 (40/70/44)	66.7/8.9/24.4 (30/4/11)	<0.001
**Passive smoking: Yes/no** (**%, *n***)	21.9/78.1 (22/122)	24.4/75.6 (11/34)	0.788
**Number of cigarettes per day: Zero/<20/≥20** (**%, *n***)	71.6/4.5/23.9 (110/7/37)	77.8/2.2/20.0 (35/1/9)	0.900
**Alcohol consumption: Yes/no** (**%, *n***)	49.0/51.0 (76/79)	84.4/15.6 (38/7)	<0.001
**Number of alcoholic drinks (10 g ethanol) per week: 0/1–4/5–10/11–20/>20 (%, n)**	51.0/21.3/13.6/10.3/3.9 (79/33/21/16/6)	15.6/43.2/20.5/13.6/6.8 (7/19/9/6/3)	0.005
**Nutritional factors**			
**Weight, kg (median (Q1–Q3))**	71.0 (60.0–81.0)	78.0 (68.7–86.5)	0.005
**BMI, kg/m^2^ (median (Q1–Q3)**	26.5 (22.9–29.6)	27.1 (24.6–30.5)	0.121
**Underweight/proper weight/overweight, obese (%, *n*)**	8.0/32.5/59.6 (12/49/90)	0.0/26.7/73.3 (0/12/33)	0.082
**Adipose tissue, % (median (Q1–Q3))**	27.8 (20.9–33.8)	33.1 (26.8–39.4)	0.004
**WHR, arbitrary unit (median (Q1–Q3))**	0.94 (0.88–0.97)	0.85 (0.80–0.93)	<0.001
**Abdominal obesity: Yes/no (%, *n*)**	77.1/22.9 (111/33)	50.0/50.0 (19/19)	0.001
**Diet energy from carbohydrate/fat/protein (%)**	51.9/30.2/15.4	50.6/28.7/16.3	
**Nutrient intake (median (Q1–Q3))**			
**Energy intake** (**kcal/day**)	2052.0 (1583.4–2546.3)	1932.5 (1560.0–2284.4)	0.277
**Protein intake** (**g/day**)	79.2 (59.7–102.5)	79.9 (68.7–98.7)	0.990
**Carbohydrate intake** (**g/day**)	285.1 (223.0–339.0)	244.8 (204.3–320.0)	0.110
**Fat intake** (**g/day**)	67.6 (50.8–91.9)	66.1 (45.6–79.2)	0.211
**Cu intake** (**mg/day**)	1.36 (1.14–1.67)	1.31 (1.10–1.46)	0.089
**Zn intake** (**mg/day**)	10.3 (8.19–13.7)	10.04 (8.64–11.69)	0.390
**Fe intake** (**mg/day**)	12.3 (9.85–15.3)	12.58 (10.0–13.64)	0.739
**Mn intake** (**mg/day**)	4.53 (3.78–5.71)	4.78 (3.15–5.94)	0.844
**Ca intake** (**mg/day**)	743.1 (565.6–970.9)	571.1 (479.3–822.0)	0.006
**Mg intake** (**mg/day**)	341.1 (290.6–413.4)	340.1 (279.7–385.3)	0.356
**Dietary fiber intake** (**g/day**)	22.5 (18.4–26.9)	20.8 (16.9–29.9)	0.578
**Clinical characteristics**			
**Clinical stage of disease: I/II/III/IV (%, *n*)**	44.9/19.1/14.7/21.3 (61/26/20/29)	NA	
**Chemotherapy: Yes (%, *n*) ***	40.7 (68)		
**Radiotherapy: Yes (%, *n*)**	10.8 (18)		
**Lung cancer resection: Yes (%, *n*)**	66.5 (111)		
**Type of lung cancer: NSCLC/SCLC/carcinoid (%, *n*)**	93.4/5.4/1.2 (156/9/2)		
**CVD: Yes/no (%, *n*)**	43.8/56.2 (67/86)		
**COPD: Yes/no (%, *n*)**	13.7/86.3 (21/132)		
**DM: Yes/no (%, *n*)**	15.0/84.9 (23/130)		
**Hb, g/dL (median (Q1–Q3))**	12.9 (11.8–14.1)		
**Anemia: Yes/no (%, *n*)**	39.1/60.9 (61/95)		
**Platelets, 10^3^ cells/µL, (median (Q1–Q3))**	254.0 (196.0–308.0)		
**Platelets <150/150–400/>400 (%, *n*)**	8.3/84.0/7.7 (13/131/12)		
**NLR, arbitrary unit (median (Q1–Q3))**	2.36 (1.61–4.97)		
**eGFR ≥ 90/<90 mL/min/1.73 m^2^ (%, *n*)**	64.3/35.7 (81/45)		
**Creatinine, mg/dL, (median (Q1–Q3))**	0.78 (0.65–0.9)		
**Creatinine <0.7/0.7–1.2/>1.2 (%, *n*)**	34.0/58.3/7.6 (49/84/11)		
**Glucose, mg/dL, (median (Q1–Q3))**	101.0 (93.2–112.2)		
**Glucose <100.0/100.0–125.9/≥126.0 (%, *n*)**	45.0/41.3/13.7 (49/45/15)		

F, female; M, male; BMI, body mass index; WHR, waist–hip ratio; NSCLC, non-small cell lung cancer; SCLC, small cell lung cancer; CVD, cardiovascular disease; COPD, chronic obstructive pulmonary disease; DM, diabetes mellitus type 1 or 2; Hb, hemoglobin; NLR, neutrophil:lymphocyte ratio; eGFR, estimated glomerular filtration rate; * percentages of post-study oncological treatment types do not sum-up to 100, due to multiple treatment regimens.

**Table 2 nutrients-13-00060-t002:** Median serum and whole blood Cu and Zn concentrations and Cu:Zn ratio in control group and lung cancer patients and in lung cancer subgroups according to different factors, presented as median (Q1–Q3).

Group	Serum Cu (mg/L)	*p*	Serum Zn (mg/L)	*p*	Serum Cu:Zn Ratio (Arbitrary Unit)	*p*	Whole Blood Cu (mg/L)	*P*	Whole Blood Zn (mg/L)	*p*	Whole Blood Cu:Zn (Arbitrary Unit)	*p*
**All study participants**												
**Lung cancer group**	1.02 (0.86–1.22)	0.327	**0.88** **(0.78–1.02)**	**<0.001**	**1.18** **(0.91–1.50)**	**<0.001**	**1.05** **(0.88–1.26)**	**<0.001**	**6.57** **(5.73–7.55)**	**<0.001**	**0.16** **(0.13–0.20)**	**<0.001**
**Control group**	0.95 (0.86–1.16)		**1.23** **(1.14–1.36)**		**0.83** **(0.71–0.94)**		**0.73** **(0.51–0.83)**		**8.79** **(7.29–9.46)**		**0.08** **(0.06–0.10)**	
**Only lung cancer patients**												
**Sex**												
**F (*n* = 76)**	**1.07** **(0.95–1.27)**	**0.007**	0.89 (0.78–1.03)	0.775	**1.23** **(1.02–1.63)**	**0.017**	1.08 (0.92–1.32)	0.083	6.36 (5.62–7.17)	0.071	**0.17** **(0.14–0.21)**	**0.009**
**M (*n* = 90)**	**0.96** **(0.81–1.19)**		0.88 (0.79–1.02)		**1.09** **(0.88–1.43)**		1.04 (0.85–1.21)		6.97 (5.87–7.72)		**0.15** **(0.11–0.18)**	
**CVD**												
**Yes (*n* = 67)**	0.96 (0.84–1.26	0.485	0.87 (0.77–1.02)	0.500	1.19 (0.89–1.45)	0.995	1.04 (0.83–1.21)	0.227	**6.98** **(6.23–7.60)**	**0.009**	**0.16** **(0.12–0.18)**	**<0.046**
**No (*n* = 85)**	1.03 (0.87–1.22)		0.88 (0.79–1.04)		1.15 (0.91–1.57)		1.08 (0.91–1.32)		**6.75** **(5.63–8.03)**		**0.17** **(0.13–0.22)**	
**COPD**												
**Yes (*n* = 21)**	0.96 (0.86–1.26)	0.872	0.88 (0.83–0.99)	0.693	1.05 (0.98–1.46)	0.769	1.00 (0.90–1.14)	0.566	**7.49** **(6.29–8.09)**	**0.022**	**0.13** **(0.12–0.17)**	**0.034**
**No (*n* = 131)**	1.00 (0.85–1.22)		0.88 (0.77–1.03)		1.18 (0.89–1.51)		1.08 (0.88–1.28)		**6.48** **(5.62–7.35)**		**0.16** **(0.13–0.20)**	
**Clinical stage of disease**												
**I (*n* = 61)**	**0.96** **(0.80–1.22) ^a^**	**0.022**	0.91 (0.78–1.06)	0.518	1.05 (0.82–1.47)	0.170	1.04 (0.88–1.24)	0.051	**6.09** **(5.44–7.10) ^a^**	**<0.004**	**0.16** **(0.13–0.21) ^ab^**	**0.013**
**II (*n* = 26)**	**1.03** **(0.85–1.24) ^ab^**		0.84 (0.75–0.95)		0.84 (0.75–0.95)		1.09 (1.00–1.27)		**6.35** **(5.30–7.06) ^a^**		**0.19** **(0.15–0.21) ^a^**	
**III (*n* = 20)**	**0.99** **(0.92–1.26) ^ab^**		0.90 (0.81–1.02)		0.90 (0.81–1.02)		0.90 (0.77–1.05)		**6.68** **(5.94–7.61) ^ab^**		**0.13** **(0.10–0.17) ^b^**	
**IV (*n* = 29)**	**1.18** **(0.93–1.49) ^b^**		0.87 (0.80–1.03)		1.34 (1.00–1.88)		1.14 (0.87–1.37)		**7.54** **(6.28–8.18) ^b^**		**0.16** **(0.10–0.21) ^ab^**	
**Hb (g/dL)**												
**<13.7 ^M^**, **<13.2 ^M#^**, **<12.2 ^F^ (*n* = 61)**	**1.17** **(0.94–1.40)**	**<0.001**	**0.82** **(0.76–0.95)**	**0.006**	**1.39** **(1.07–1.71)**	**<0.001**	1.05 (0.91–1.26)	0.561	6.56 (5.90–7.55)	0.765	0.16 (0.13–0.21)	0.367
**≥13.7 ^M^**, **≥13.2 ^M#^**, **≥12.2 ^F^ (*n* = 95)**	**1.13** **(1.01–1.35)**		**0.91** **(0.81–1.06)**		**1.06** **(0.82–1.33)**		1.06 (0.86–1.29)		6.48 (5.61–7.48)		0.16 (0.13–0.20)	
**Platelets (10^3^ cells/µL)**												
**<150 (*n* = 12)**	**0.87** **(0.78–1.18) ^a^**	**0.012**	0.86 (0.74–1.02)	0.606	1.18 (0.78–1.35)	0.352	0.89 (0.77–1.32)	0.431	5.97 (5.52–7.24)	0.763	0.16 (0.13–0.18)	0.701
**150**–**400 (*n* = 131)**	**1.00** **(0.85–1.21) ^a^**		0.88 (0.79–1.02)		1.15 (0.90–1.51)		1.07 (0.88–1.28)		6.57 (5.87–7.49)		0.16 (0.13–0.20)	
**>400 (*n* = 13)**	**1.41** **(1.06–1.49) ^b^**		0.89 (0.80–1.08)		1.39(1.05–1.89)		1.08 (0.91–1.28)		6.25 (5.63–7.69)		0.16 (0.13–0.23)	
**Glucose (mg/dL)**												
**<100 (*n* = 49)**	0.98 (0.84–1.22)	0.579	0.91 (0.81–1.09)	0.887	1.10 (0.83–1.45)	0.600	1.04 (0.86–1.33)	0.573	**6.37** **(5.33–7.37) ^a^**	**0.020**	0.17 (0.13–0.23)	0.634
**100**–**125 (*n* = 45)**	1.04 (0.93–1.21)		0.88 (0.75–1.04)		1.18 (0.93–1.59)		1.10 (0.83–1.32)		**6.40** **(5.72–7.63) ^ab^**		0.16 (0.12–0.21)	
**>125 (*n* = 15)**	1.19 (0.85–1.41)		0.89 (0.84–1.41)		1.33 (0.92–1.65)		1.15 (0.94–1.31)		**7.54** **(6.57–9.13) ^b^**		0.14 (0.11–0.17)	
**Ca intake (mg/day)**												
**<EAR**	**1.04** **(0.90–1.27)**	**0.045**	0.90 (0.80–1.03)	0.231	1.20 (0.93–1.57)	0.344	1.06 (0.92–1.30)	0.122	6.57 (5.61–7.43)	0.525	0.17 (0.13–0.21)	
**≥EAR**	**0.98** **(0.82–1.19)**		0.86 (0.76–1.01)		1.14 (0.89–1.44)		1.04 (0.79–1.23)		6.56 (5.87–7.66)		0.15 (0.12–0.18)	

F, female; M, male; DM, diabetes mellitus type 1 or 2; CVD, cardiovascular disease; COPD, chronic obstructive pulmonary disease; Hb, hemoglobin, # indicates men over 60 years old; EAR, estimated average requirement; *p* < 0.05, values with different superscript letters are significantly different; significant differences are bolded.

**Table 3 nutrients-13-00060-t003:** Univariate Cox regression models predicting lung cancer mortality.

Risk Factor	Univariate Cox Regression Models		
	HR	95% CI HR	*p*
**Smoking vs**. **nonsmoking**	1.81	1.00–3.29	0.049
**Clinical stage III vs**. **I**	2.31	1.15–4.64	0.019
**Clinical stage IV vs**. **I**	4.52	2.53–8.11	<0.001
**NLR >2.36 vs**. **<2.36 arbitrary unit**	1.91	1.14–3.20	0.013
**Glucose >125 vs**. **<100 mg/dL**	2.25	1.13–4.50	0.021
**Platelets >400** × **10^3^ vs**. **150**–**400** × **10^3^ cells/µL**	2.13	1.09–4.17	0.027
**Serum Cu per 0.1 mg/L**	1.18	1.10–1.27	<0.001
**Serum Cu:Zn ratio per 0.1 arbitrary unit**	1.06	1.02–1.11	<0.008
**Whole blood Zn per 1 mg/L**	1.32	1.15–1.51	<0.001

NLR, neutrophil lymphocyte ratio; HR, hazard ratio; CI, confidence interval.

**Table 4 nutrients-13-00060-t004:** Multiple Cox regression models predicting lung cancer mortality.

Model	Parameters	HR	95% CI HR	*p*
**I (for serum Cu)**	Serum Cu per 0.1 mg/L	1.13	1.03–1.24	0.010
	Clinical stage IV vs. I	3.14	1.43–6.89	0.004
	Clinical stage III vs. I	3.25	1.35–7.78	0.008
	Clinical stage II vs. I	0.75	0.26–2.20	0.604
**II (for serum Zn)**	Clinical stage IV vs. I	3.72	1.73–8.03	<0.001
	Clinical stage III vs. I	3.52	1.48–8.37	0.004
	Clinical stage II vs. I	0.89	0.31–2.59	0.842
**III (for serum Cu:Zn ratio)**	Serum Cu:Zn per 0.1 arbitrary unit	1.06	1.01–1.11	0.009
	DM: yes vs. no	1.83	1.00–3.35	0.049
**IV (for whole blood Cu)**	Whole blood Cu per 0.1 mg/L	1.09	1.00–1.19	0.04
	Clinical stage IV vs. I	3.24	1.49–7.06	0.003
	Clinical stage III vs. I	3.81	1.52–9.49	0.004
	Clinical stage II vs. I	0.77	0.27–2.25	0.639
**V (for whole blood Zn)**	Whole blood Zn per 1 mg/L	1.25	1.04–1.52	0.021
	Clinical stage IV vs. I	2.59	1.13–5.94	0.024
	Clinical stage III vs. I	2.99	1.26–7.14	0.013
	Clinical stage II vs. I	0.69	0.24–2.01	0.496
**VI (for whole blood Cu:Zn)**	Clinical stage IV vs. I	3.54	1.65–7.59	0.001
	Clinical stage III vs. I	3.12	1.28–7.57	0.012
	Clinical stage II vs. I	0.91	0.31–2.61	0.853

DM, diabetes mellitus type 1 or 2. Models I, II, and IV–VI were adjusted by the following parameters: Age (continuous), sex, passive smoking (yes/no), smoking status (yes/no/previous), alcohol consumption (yes/no), BMI (<20/20–24.9/>24.9 kg/m^2^), abdominal obesity, DM (yes/no), CVD (yes/no), COPD (yes/no), clinical stage (I, II, III, IV), anemia (yes/no), platelet count (<150/150–400/>400 × 10^3^ cells/µL), and NLR (<2.36 vs. >2.36 arbitrary unit). Model III was adjusted by age (continuous), sex, passive smoking (yes/no), smoking status (yes/no/previous), alcohol consumption (yes/no), BMI (<20/20–24.9/>24.9 kg/m^2^), DM (yes/no), CVD (yes/no), COPD (yes/no), anemia (yes/no), and platelet count (<150/150–400/>400 × 10^3^ cells/µL).

**Table 5 nutrients-13-00060-t005:** Follow-up times and events of all-cause death in lung cancer patients by serum Cu, whole blood Zn concentrations, and serum Cu:Zn ratio.

Parameters	Person-Months	Number of Events	Incidence Rates *	Median (Range) of Follow-up Time
Overall	6929.26	83	11.98	43.04 (0.23–85.81)
Serum Cu < median	3903.98	33	8.45	51.00 (0.23–79.40)
Serum Cu > median	3025.28	50	16.53	34.95 (0.98–85.80)
Serum Cu:Zn < median	3972.40	36	9.06	55.79 (0.23–80.52)
Serum Cu:Zn > median	2953.28	46	15.58	44.45 (0.99–85.81)
Whole blood Zn < median	3659.15	35	9.56	51.55 (1.05–85.81)
Whole blood Zn > median	2970.31	47	15.82	43.46 (0.23–78.71)

* per 1000 person-months; event (endpoint) was defined as all-cause mortality in lung cancer patients.

## Data Availability

The data presented in this study are available on request from the corresponding author. The data are not publicly available.

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
