# Peer review of "Serum and Whole Blood Cu and Zn Status in Predicting Mortality in Lung Cancer Patients"

_nutrients, 2020, doi:10.3390/nu13010060_

Round 1
Reviewer 1 Report
This paper is of interest for Nutrients readership but have few points that have to be addressed.
1) Page 8. The authors say "whole blood Cu levels were significantly increased in lung cancer 213 patients compared to control subjects". It should be discussed whether this copper is ceruloplasmin bound or not. Ceruloplasmin-bound copper is normally used by tissues and organs, while so called "ceruloplasmin-free" copper has a higher potential to induce the oxidative damage.
2) Page 14. Along the same line, the authors note that "abnormal circulating Cu and Zn concentration may contribute to cancer development by dysregulation of redox enzyme activity, therefore DNA damage and protein modification". This statement is very generic. Higher copper levels should not necessarily cause a dysregulation/damage. Tumors just require higher copper levels to sustain rapid growth, which in turn needs extensive copper supply to mitochondrial COX for respiration and to LOX and similar enzymes for metastatic expansion (see recent 2020 review from Petris lab).
3) Page 15. The authors note that copper might be needed for angiogenesis. I think that this should be expanded and copper-dependent components of angiogenic machinery has to be at least mentioned (see review from Fukai et al., 2018)
4) The authors also discuss anemia prevalence in patients with elevated blood copper levels. Here it would be probably worth again discussing regarding ceruloplasmin. Ceruloplasmin is a ferroxidase and is involved in iron metabolism. Its lack leads to anemia. So it is pempting to speculate that the blood copper increase in patients is not due to increase in ceruloplasmin. I think that this point might be discussed.
Author Response
Thank you very much for all your constructive comments concerning our manuscript. We have improved the text and included additional data in accordance with all your suggestions. All changes are marked in red. A detailed description of changes in the manuscript appears in the list below.
- Page 8. The authors say "whole blood Cu levels were significantly increased in lung cancer 213 patients compared to control subjects". It should be discussed whether this copper is ceruloplasmin bound or not. Ceruloplasmin-bound copper is normally used by tissues and organs, while so called "ceruloplasmin-free" copper has a higher potential to induce the oxidative damage.
Dear Reviewer, thank you very much for this relevant comment. In fact, we analyze absolute serum and whole blood copper concentration, therefore based on these results, we are not able to separate the fraction of ceruloplasmin-bound copper and ceruloplasmin-free copper. Unfortunately, we did not perform analyzes on serum ceruloplasmin concentration. The aim of our work was to evaluate circulating Cu and Zn concentration as predictive markers, but we fully agree that such an indirect speciation of circulating Cu would help to comprehend the relevance of the observed links between the microelement status and disease endpoint. Due to lack of information on fractions of Cu, we put this limitation into limitations of the study. Moreover, we discussed this limitation in the discussion section. Additional information are in Discuss section: Further studies are needed to more precisely evaluate disturbances in circulating Cu in lung cancer, mainly via assessment the relative levels of free- and ceruloplasmin-bound copper and their influence on cancerogenesis and mortality. Unbound-Cu is responsible for free radical reactions e.g. Haber-Weiss reaction, and thus lead to alterations in functions of key biomolecules as protein, lipids, and nucleic acids (line: 379-383), and in limitation of the study: Moreover, we did not perform measurements of ceruloplasmin concentration, therefore we were not able to assess ceruloplasmin-bound and ceruloplasmin-free copper. This information is warranted for further evaluation of mechanisms related to associations between copper status and risk of mortality in lung cancer patients as well as cancer-related anemia (line: 500-501).
- Page 14. Along the same line, the authors note that "abnormal circulating Cu and Zn concentration may contribute to cancer development by dysregulation of redox enzyme activity,therefore DNA damage and protein modification". This statement is very generic. Higher copper levels should not necessarily cause a dysregulation/damage. Tumors just require higher copper levels to sustain rapid growth, which in turn needs extensive copper supply to mitochondrial COX for respiration and to LOX and similar enzymes for metastatic expansion (see recent 2020 review from Petris lab).
Dear Reviewer, once again thank you very much for pointing out this weakness of our discussion. We have improved the discussion on the role of copper in cancerogenesis with the mechanism of Cu activity indicated in recent manuscript prepared by Shanbhad et al. (2020). Additional paragraph is in Discuss section: On the other hand, higher Cu concentration is needed to meet the metabolic demands of the growth and development of cancer cells. Cancer cells require higher copper resources for e.g. the Warburg effect – an early metabolic switch by which cancer cells produce energy via glycolysis instead of mitochondrial oxidative phosphorylation. Moreover, increased Cu concentration is needed to potentiate the activity of cupro-enzymes like lysyl oxidase (LOX), superoxide dismutase (SOD) and cytochrome-c-oxidase (COX) as well as some molecular pathways which support cancer cell proliferation and survival, like mitogen-activated protein (MAP) kinases and autophagic ULK1/2 kinases [15] (line 388 - 395).
- Page 15. The authors note that copper might be needed for angiogenesis. I think that this should be expanded and copper-dependent components of angiogenic machinery has to be at least mentioned (see review from Fukai et al., 2018)
Dear Reviewer, thank you very much for this comment and for bringing our attention to this valuable manuscript. We added information regarding copper-dependent components playing a role in angiogenesis, based on Fukai et al.(2018) review. Additional information are in Discussion section: It has been shown that expression of VEGF, one of the crucial components of tumor angiogenesis, is Cu-mediated, partly by activation of hypoxia-inducible factor 1-α [42,43]. Moreover, FGF1, angiogenic growth factor, is a Cu-binding protein and its secretion requires the Cu-dependent formation of a complex. Cu also potentiate the affinity with which angiogenin, binds to endothelial receptors. Additionally, antioxidant-1 (Atox-1), cytosolic protein, responsible for transferring Cu(I) to the Cu-ATPases, in inflammatory neovascularization via regulating ROS/NF-κB pathway (line: 419 - 426).
- The authors also discuss anemia prevalence in patients with elevated blood copper levels. Here it would be probably worth again discussing regarding ceruloplasmin. Ceruloplasmin is a ferroxidase and is involved in iron metabolism. Its lack leads to anemia. So it is pempting to speculate that the blood copper increase in patients is not due to increase in ceruloplasmin. I think that this point might be discussed.
Dear Reviewer, thank you very much for this additional comment. As we did not measure ceruloplasmin concentration we could not interprete our results on anemia in regard to ceruloplasmin concetration. However, we made an effort to discuss it: Therefore, the positive relationship between circulating Cu concentration and anemia in this study probably mainly resulted from a common link, the inflammatory process, and often observed in lung cancer patients ceruloplasmin alterations. Ceruloplasmin plays a crucial role in iron release from cells and tissue into circulatory system. Patients with ceruloplasmin deficiency may accumulate iron in tissue and therefore their serum iron concentration is decreased [53]. However, we did not evaluate serum ceruloplasmin concentration in this study, therefore alterations in ceruloplasmin that are mentioned above remain only speculative, and requires further research (line: 460 - 466). Additionally, as we mentioned above, we put this limitation of lack of ceruloplasmin determination in our study into Discussion section: Moreover, we did not perform measurements of ceruloplasmin concentration, therefore we were not able to assess ceruloplasmin-bound and ceruloplasmin-free copper. This information is warranted for further evaluation of mechanisms related to associations between copper status and risk of mortality in lung cancer patients as well as lung cancer-related anemia (line 500 - 503).
Dear Reviewer, once again thank you very much for all your valuable comments. We hope that these modifications of our manuscript meet your approval.
Reviewer 2 Report
This paper report data regarding a prospective cohort study aimed to evaluate the relationships between serum and whole blood copper and zinc status and clinical, sociodemographic, and nutritional data, and to investigate the association of copper and zinc status with all-cause mortality in lung cancer.
Trace elements are essential micronutrients which are involved in several biological mechanisms taking functions as cofactors for the activity of antioxidant enzymes, cell division and differentiation.
Although the topic is interesting, the paper has several weaknesses. Among the most relevant:
- More detail should be provided on study design and endpoints.
- The authors should provide inclusion and exclusion criteria.
- Had the authors informed consent for this study?
- The authors should describe dietary intake evaluation methods clearly. This is vague. The level of this intervention may be different due to each operator's skill.
- The authors evaluated body fat percentage and waist-hip ratio, why in this context? Particularly, the percentage of body fat was assessed using a body fat monitor. Please, clarify the scientific rationale in the text.
- A definition of abdominal obesity was not reported.
- The overall results section would benefit from revision in order to make it easier to understand the main findings.
- Clinical and pathological characteristics of lung cancer patients should be better reported.
- Post-study therapies in lung cancer patients should be provided.
- Median follow-up is not reported.
- The time to event analyses should be better reported in tables with the number of events and person/time of follow-up for lung cancer patients.
- The number at risk is missing from Fig 1-2 and 3.
- Actually, Cu/Zn ratio was shown to be associated with nutritional patterns, oxidative stress and immune abnormalities. However, it was not possible to control all factors influencing Cu and Zn status such as diet and lifestyle. Please, develop more this section.
- The present study has several limitations. Particularly, there is a risk of several residual confounding. This, I suggest the authors to be more careful in the conclusion: Circulatory status of Cu and Zn should be involved in routine clinical characteristics of patients with lung cancer as an additional prognostic variables.
- I recommend the authors work with an English-speaking editor to review the paper for grammar and punctuation.
Author Response
Thank you very much for all your constructive comments concerning our manuscript. We have improved the text and included additional data in accordance with all your suggestions. All changes are marked in red. A detailed description of changes in the manuscript appears in the list below.
This paper report data regarding a prospective cohort study aimed to evaluate the relationships between serum and whole blood copper and zinc status and clinical, sociodemographic, and nutritional data, and to investigate the association of copper and zinc status with all-cause mortality in lung cancer.
Trace elements are essential micronutrients which are involved in several biological mechanisms taking functions as cofactors for the activity of antioxidant enzymes, cell division and differentiation.
Although the topic is interesting, the paper has several weaknesses. Among the most relevant:
- More detail should be provided on study design and endpoints.
Dear Reviewer, thank you very much for pointing out this weakness. We added additional, more detailed information on study design and endpoints. An additional subsection of Materials and Methods is added in revised version: This study was designed to evaluate following hypotheses: (i) circulating (serum and whole blood) Cu and Zn status is disturbed in lung cancer patients (ii) this alteration is related to following parameters; clinical: stage of disease, prevalence of co-morbidities like cardiovascular disease (CVD), chronic obstructive pulmonary disease (COPD), diabetes mellitus type 1 or 2 (DM 1 or 2), anemia prevalence, alterations in hemoglobin (Hb), platelets, serum creatinine, glucose concentration, estimated globular filtration (eGFR) and neutrophil:lymphocyte ratio (NLR); nutritional: nutritional status (with emphasis on overweight and obesity prevalence), dietary intakes of total energy, macronutrients and microelements; sociodemographic: sex, age, level of education, active and passive smoking status as well as alcohol consumption, and (iii) serum and whole blood Cu and Zn can be used to predict all-cause lung cancer death (endpoint), since alterations in circulatory Cu and Zn status as well as above-mentioned factors related to clinical, nutritional and sociodemographic data influence all-cause mortality among lung cancer patients (line: 73 - 84).
- The authors should provide inclusion and exclusion criteria.
We added inclusion and exclusion criteria of this study into the Study subjects subsection: Inclusion criteria for lung cancer patients were as follows: confirmed diagnosis of primary lung cancer, lack of mental disease which could make difficult to obtain reliable answers, no other primary malignant diseases (line: 89 - 91). Exclusion criteria for lung cancer patients were as follows: unconfirmed diagnosis of lung cancer, lung cancer as metastasis of other primary cancer, prior chemotherapy, radiotherapy or other oncological therapy for any malignant disease (line: 97 - 100).
- Had the authors informed consent for this study?
This study was approved by the Ethics Commission of Wroclaw Medical University (approval no. 540/2013), and the study was conducted according to the principles expressed in the Declaration of Helsinki. All participants provided written consent to take part in the research. This information is also included in manuscript body (line: 106 - 108).
- The authors should describe dietary intake evaluation methods clearly. This is vague. The level of this intervention may be different due to each operator's skill.
Thank you very much for this comment, we added more detailed information on method regarding dietary assessment, we hope that this information clarify the procedure. Please, see Anthropometric measurements and dietary intake subsection, (line: 130-152)
- The authors evaluated body fat percentage and waist-hip ratio, why in this context? Particularly, the percentage of body fat was assessed using a body fat monitor. Please, clarify the scientific rationale in the text.
We decided to evaluate nutritional status with baseline anthropometric measurements due to important influence of nutritional status on body Zn, Cu status. Because majority of our lung cancer participants were overweight or obesity (based on BMI - ca. 60%), during design of the study, we decided to include also this data to present any association with circulating Cu and Zn status. Especially, due to reported in latest study performed by Rios-Lugo et al. (2020) and meta-analysis performed by Gu et al. (2020) that overweight or obesity may be independent factors influencing homeostasis of Cu and Zn. However, baseline statistical analyses did not reveal any such relationships: BMI (<20 vs 20-24.9 vs. >25 kg/m2) and WHR (abdominal obesity: yes vs. no) did not influence serum and whole blood concentrations of Cu and Zn and the Cu:Zn ratio among lung cancer patients and these results are included in supplementary data. This explanatory statement is added to the subsection Due to significant impact of nutritional status on circulating zinc and copper status, we performed baseline anthropometric measurements to evaluate this associations among participants enrolled to this study, with emphasis to overweight and obesity [22,23] (line: 130-132).
- A definition of abdominal obesity was not reported.
Dear Reviewer, thank you very much for pointing out this missing information. Now, the definition of abdominal obesity is reported in the revised version: Abdominal obesity was determined based on WHR, and following WHO criteria: ≥0.90 for men, ≥0.85 for women [24]. (line: 138 - 139).
- The overall results section would benefit from revision in order to make it easier to understand the main findings.
We added subsection titles (line: 226, 255, and 293) in Result section to easier follow the main findings. The subsection titles are in line with the main hypotheses, that were tested. We hope that this modification improves the readability of Result section.
- Clinical and pathological characteristics of lung cancer patients should be better reported.
We have improved clinical and pathological characteristic of lung cancer patients, please see – line: 235 – 249.
- Post-study therapies in lung cancer patients should be provided.
Dear Reviewer, post-study therapies are presented in Table 1 as well as are mentioned in Result section (line: 237 – 239).
- Median follow-up is not reported.
In revised version, we reported median (range) follow up of time for the study (line: 194).
- The time to event analyses should be better reported in tables with the number of events and person/time of follow-up for lung cancer patients.
Thank you for this valuable suggestion improving our results. The table with number of events, person-months of follow up, median (range) of follow up and incidence rates per 1000 person-months are presented in Table 5 (line: 354 – 356) and mentioned in Result section: Follow up times and events of all-cause cancer death in lung cancer patients by serum Cu, whole blood Zn and serum Cu:Zn ratio are presented in Table 5. Serum Cu concentration above median displayed the highest incidence rate: 16.53 per 1000 person-months, while serum Cu below median - the lowest incidence rate with 8.45 per 1000 person months (line 347 – 351).
- The number at risk is missing from Fig 1-2 and 3.
We have added number at risk on Figures 1-3 as well as on Figure S1.
- Actually, Cu/Zn ratio was shown to be associated with nutritional patterns, oxidative stress and immune abnormalities. However, it was not possible to control all factors influencing Cu and Zn status such as diet and lifestyle. Please, develop more this section.
Thank you very much for pointing out this issue relevant to conclusion of our study. We included additional paragraph into limitation of our study, where we indicate directions of further studies that are needed in careful assessment of predictive value of Cu and Zn status in lung cancer patients: Actually, circulating Cu and Zn status has been shown to be associated with dietary intakes, disturbances in redox status, immune system, and metabolic derangements [12,15,46,63]. In this study, it was however not possible to control all factors influencing Cu and Zn status such as diet, lifestyle, and clinical parameters, therefore further studies should be performed in a more homogeneity group of lung cancer patients, or analyses should be adjusted also for the mentioned-above data to better associate the impact of circulating Cu and Zn status on predicting mortality in lung cancer patients (line: 507 - 513).
- The present study has several limitations. Particularly, there is a risk of several residual confounding. This, I suggest the authors to be more careful in the conclusion: Circulatory status of Cu and Zn should be involved in routine clinical characteristics of patients with lung cancer as an additional prognostic variables.
Thank you very much for your valuable insights on our conclusions. We have revised them in accordance with your suggestions. Now, they are as follows: In conclusion, higher serum Cu level, Cu:Zn ratio and whole blood Zn level was shown to negatively affect all-cause mortality risk in lung cancer patients. However, further, more detailed and with larger cohort, studies on the role of Cu and Zn status in predicting all-cause and cancer-cause mortality among lung cancer patients as well as studies on mechanisms underpinning surprising negative prediction of lung cancer survival in patients with elevated whole blood Zn status are warranted. This could be helpful in developing strategies to improve outcome of patients with regard to the body status of these elements. Findings of the study indicate that circulatory status of Cu and Zn might be included in routine clinical characteristics of patients with lung cancer patients as an additional prognostic variables, but only after further mentioned above studies (line: 515 - 523).
- I recommend the authors work with an English-speaking editor to review the paper for grammar and punctuation.
We have reviewed the paper for grammar and punctuation. Additionally, the manuscript underwent English language editing by MDPI.
Dear Reviewer, once again thank you very much for all your comments improving this manuscript. We hope that these modifications of our manuscript meet your approval.

Reviewer 3 Report
The paper has been well written and provides important messages.
- What is the cut-off value of serum Cu concentration, blood Zn concentration and serum Cu:Zn ratio between survival group and non-survival group in lung cancer patients ?
- As shown in table 2, serum Cu and Zn level were increased in stage III and stage IV lung cancer (Advanced). Is it possible to show the cut-off value of Cu and Zn level between early and advanced lung cancer?
Author Response
Thank you very much for all your constructive comments concerning our manuscript. We have improved the text and included additional data in accordance with all your suggestions. All changes are marked in red. A detailed description of changes in the manuscript appears in the list below.
Reviewer #3
- The paper has been well written and provides important messages.
Dear Reviewer, thank you very much for your positive review of our manuscript and for suggestions of presenting additional, useful results. We have addressed each of your comments below.
- What is the cut-off value of serum Cu concentration, blood Zn concentration and serum Cu:Zn ratio between survival group and non-survival group in lung cancer patients ?
In result section we presented additional data on ROC analysis, which allow us to evaluate cut-off value of serum Cu concentration, whole blood Zn concentration, and serum Cu:Zn ratio between survival and non-survival patients enrolled in this study: We estimated cut-off values: 1.19 mg/L for serum Cu concentration, 1.34 for serum Cu:Zn ratio and 7.54 mg/L for whole blood Zn concentration, which best differentiated survival from non-survival lung cancer patients. Areas under the ROC curve (AUC, 95% CI) for serum Cu concentration, Cu:Zn ratio, and whole blood Zn concentration were as follows: 0.668 (0.586 – 0.750), 0.604 (0.518 – 0.690) and 0.637 (0.551 – 0.723) (Supplementary Table S2) (line: 343 - 347).
- As shown in table 2, serum Cu and Zn level were increased in stage III and stage IV lung cancer (Advanced). Is it possible to show the cut-off value of Cu and Zn level between early and advanced lung cancer?
Dear Reviewer, according to your suggestion, we estimated cut-off values for serum Cu and whole blood Zn concentration between lung cancer. We decided to compare I-III stages vs IV, because metastatic stage significantly differentiates treatment options and prognosis in lung cancer. Moreover, in such division, we were observed higher AUC of ROC curves than when comparing I-II vs III-IV. Cut-off values estimated in this way are presented in Supplementary Table S2 and are mentioned in Results section:
Cut-off values of serum Cu and whole blood Zn concentrations that best differentiated patients with clinical stage IV vs clinical stage I-III were as follows: 1.42 mg/L and 7.12 mg/L, with following AUCs (95% CI): 0.658 (0.537 – 0.779) and 0.705 (0.599 – 0.811), respectively (both p<0.05) (Supplementary Table S2) (line: 270 - 274).
Thank you very much for the suggestion of setting the cut-off of Cu and Zn level between early and advanced lung cancer, and we hope that the combination of groups used for this comparison will be accepted by the Reviewer.
Round 2
Reviewer 2 Report
I am satisfied with the last one